# Flaxseed Ethanol Extract Effect in Acute Experimental Inflammation

**DOI:** 10.3390/medicina58050582

**Published:** 2022-04-24

**Authors:** Elisabeta Ioana Chera, Tiberia Ioana Pop, Raluca Maria Pop, Marcel Pârvu, Ana Uifălean, Florinela Adriana Cătoi, Andra Diana Cecan, Camelia Manuela Mîrza, Patriciu Achimaș-Cadariu, Alina Elena Pârvu

**Affiliations:** 1Department of Pathophysiology, Iuliu Hațieganu University of Medicine and Pharmacy, 400012 Cluj-Napoca, Romania; elisabeta.chera@yahoo.com (E.I.C.); uifaleanana@gmail.com (A.U.); florinela12@yahoo.com (F.A.C.); andra.cecan@umfcluj.com (A.D.C.); camelia.mirza@umfcluj.ro (C.M.M.); parvualinaelena@umfcluj.ro (A.E.P.); 2Department of Technical and Soil Sciences, Faculty of Agriculture, University of Agricultural Sciences and Veterinary Medicine, 400012 Cluj-Napoca, Romania; 3Department of Pharmacology, Toxicology and Clinical Pharmacology, Iuliu Haţieganu University of Medicine and Pharmacy, 400012 Cluj-Napoca, Romania; 4Faculty of Biology and Geology, Babeș-Bolyai University, 400012 Cluj-Napoca, Romania; marcel.parvu@ubbcluj.ro; 5Department of Oncology, University of Medicine and Pharmacy Iuliu Hațieganu, 400012 Cluj-Napoca, Romania; pachimas@umfcluj.ro

**Keywords:** flaxseed, antioxidant, anti-inflammatory, prophylaxic, therapeutic

## Abstract

*Background and Objectives*: Previous studies demonstrated antioxidant activities for flaxseed and flaxseed oil. The aim of the present study was to evaluate the prophylactic and therapeutic anti-inflammatory and antioxidant effects of flaxseed ethanol extract in acute experimental inflammation. *Materials and Methods*: The in vivo anti-inflammatory and antioxidant activity was evaluated on a turpentine-induced acute inflammation (6 mL/kg BW, i.m.) by measuring serum total oxidative status, total antioxidant reactivity, oxidative stress index, malondialdehyde, total thiols, total nitrites, 3-nitrotyrosine, and NFkB. The experiment was performed on nine groups (n = 5) of male rats: negative control; inflammation; three groups with seven days of flaxseed extract (100%, 50%, 25%) pretreatment followed by inflammation on day eight; three groups of inflammation followed by seven days of treatment with flaxseed extract (100%, 50%, 25%); inflammation followed by seven days of treatment with diclofenac (20 mg/kg BW). *Results*: Flaxseed extract anti-inflammatory activity was better in the therapeutic plan than in the prophylactic one, and consisted of NO, 3NT, and NF-κB reduction in a dose dependent way. ROS was reduced better in the therapeutic flaxseed extracts administration, and antioxidants were increased by the prophylactic flaxseed extracts administration. Both, ROS and antioxidants were influenced more by the total flaxseed extract, which was also more efficient than diclofenac. *Conclusions*: flaxseed extract prophylaxis has a useful antioxidant activity by increasing the antioxidants, and flaxseed extract therapy has anti-inflammatory and antioxidant activities by reducing NF-κB, RNS, and ROS.

## 1. Introduction

Inflammation is a complex reaction in response to exogenous and endogenous stimuli, that tries to remove both the cause and the consequences of cell injury. Recent observations suggested that the resolution phase is not the end of the inflammatory response, and there is a further immunological activity that leads to an adapted homeostasis, which is different from the state before the inflammation. Moreover, it was suggested that this can be a risk for the development of chronic inflammation [1], because, in many chronic conditions, the inflammatory response continues and leads to significant tissue/organ damage and abnormal repair/remodeling [2]. It is known that excessive or chronic inflammatory response is involved in the pathogenesis of many chronic diseases. In pathological inflammation professional phagocytic cells and nonphagocytic cells produce excessive ROS/RNS which diffuse in the extracellular space and induce local oxidative stress and tissue injury [3]. Thus, inflammation and oxidative stress are tightly linked pathophysiological processes, simultaneously found in the pathogenesis of many chronic diseases. Therefore, the treatment has to target both inflammation and oxidative stress, to use both antioxidants and anti-inflammatory substances simultaneously [4].

Many plant products have anti-inflammatory and antioxidant activity. Phytochemicals are considered important potential anti-inflammatory compounds because the conventional anti-inflammatory drugs currently in use treat acute inflammatory diseases but fail to treat some chronic inflammatory diseases. Natural antioxidants are considered to be safe and an alternative to currently available synthetic antioxidants with important side effects [5]. Even if many plants are already used in drugs preparation and are consumed as food due to their antioxidant effects, today herbal medicines seem to be an underutilized tool against diseases. That makes the search for new natural antioxidants of plant origin that can prevent and/or treat diseases become an important issue [6].

Flaxseeds were used in folk medicine from ancient times due to the protective effects in some chronic diseases, like cancer and cardiovascular and inflammatory diseases [7]. Today, flaxseeds are mainly used for diet due to the health benefits associated with the high content in alpha linoleic-acid (ALA), dietary fiber, and lignans [8,9]. It was mentioned that together lignans and ALA mediate anti-inflammatory activities [7]. Therefore, the present study investigated the preventive and therapeutic anti-inflammatory and antioxidant potential of flaxseed extract in an experimental turpentine oil induced acute inflammation.

## 2. Materials and Methods

### 2.1. Chemicals

Methanol, vanadium chloride (III) (VCl3), sulfanylamide (SULF), N- (1-Naphthyl) ethylenediaminedihydrochloric acid (NEDD), ferrous ammonium sulfate, xylenol orange [ocresosulfonphthalein-3,3-bis (sodium methyliminodiacetate)], orthodianisidinedihydrochloric acid (3-3′-dimethoxybenzidine), hydrogen peroxide (H_2_O_2_), sulfuric acid, hydrochloric acid, thiobarbituric acid, o-phthalaldehydewere purchased from Merck (Darmstadt, Germany); Trolox (6-hydroxy-2,5,7,8-tetramethylchroman-2-carboxylic acid) were purchased from Alfa-Aesar (Karlsruhe, Germany). The 3-nitrotyrosine ELISA kit was purchased from ABNOVA (KA0445-ABNOVA EMBLEM, Heidelberg, Germany), and the nuclear factor-κB (NFkB) ELISA kit from Biothec (EU2560- Fine Biothec, Wuhan, China).

### 2.2. Plant Extract Preparation

Grounded flaxseeds (Sanovita SRL, Ulmetel, Romania) screened through a 200 μm Retsch sieve were used for flaxseeds ethanolic extract (FLAX) preparation. Shortly, by using a modified Squibb repercolation method, 50 g of flaxseeds flour were extracted with 70% ethanol (1/1.2 g/mL) [10].

### 2.3. Experimental Design

#### 2.3.1. Animal Subjects

Wistar albino rats (strain Crl:WI) obtained from the Animal Centre (IuliuHatieganu University of Medicine and Pharmacy Cluj-Napoca), with a mean weight of 250 (±15) g, were used to evaluate the effect of FLAX on the acute experimental inflammation. The animals were housed in polypropylene cages, with free access to standard pellet food and water, and a proper laboratory environment (12 h light/dark cycles, at an ambient temperature of 21 °C). The experiments were approved by the Ethics Committee from both “IuliuHațieganu” University of Medicine and Pharmacy Cluj-Napoca and the Romanian National Sanitary Veterinary and Food Safety Authority (nr. 168/06.06.2019). All in vivo experiments were conducted in triplicate.

#### 2.3.2. Experimental Protocol

Acute inflammation was induced by one administration of turpentine oil (6 mL/kg BW, i.m.). For the evaluation of the prophylactic and therapeutic FLAX anti-inflammatory and antioxidant activities, 9 groups (n = 5) were used: (1) negative control (CONTROL), with no inflammation and no treatment; (2) inflammation (INFLAM), with no treatment; (3) seven days of treatment with FLAX 100% (1 mL/200 g BW) followed by inflammation in day 8 (FLAX100/INFLAM); (4) seven days of treatment with FLAX 50% (1 mL/200 g BW) followed by inflammation in day 8 (FLAX50/INFLAM); (5) seven days of treatment with FLAX 25% (1 mL/200 g BW) followed by inflammation in day 8 (FLAX25/INFLAM); (6) inflammation followed by seven days of treatment with FLAX 100% (1 mL/200 g BW) (INFLAM/FLAX100); (7) inflammation followed by seven days of treatment with FLAX 50% (1 mL/200 g BW) (INFLAM/FLAX50); (8) inflammation followed by seven days of treatment with FLAX 25% (1 mL/200 g BW) (INFLAM/FLAX25); (9) inflammation followed by seven days of treatment with diclofenac (20 mg/kg BW) (INFLAM/DICLO). All prophylactic and therapeutic treatments were performed by gavage (1 mL/day p.o.). On day 9, animals were anesthetized (20 mg/kg BW xylazine, 50 mg/kg BW ketamine), blood samples were collected by retro-orbital puncture and the obtained serum was kept at −80 °C until analysis [10]. Afterward, the animals were sacrificed by cervical dislocation.

#### 2.3.3. In Vivo Anti-inflammatory and Oxidative Stress Markers

The anti-inflammatory activity was assessed by measuring serum Nuclear factor-κB (NF-κB), 3-nitrotyrosine (3NT), and total nitrates and nitrates (NOx). NF-κB and 3NT were measured according to the ELISA kits instructions. The Griess reaction was used to indirectly determine NOx and the results were expressed as nitrite µmol/L. The serum oxidative stress markers were total antioxidant reactivity (TAR), total oxidative status (TOS), oxidative stress index (OSI), total thiols (SH), and malondialdehyde (MDA) [10,11] by using colorimetric assays. TAR was results are expressed as mmolTroloxequiv/L [12], and TOS results are expressed in µmol H_2_O_2_equiv/L [13]. OSI, an index of total oxidative stress, was calculated: OSI (arbitrary unit) = TOS (mol H_2_O_2_equiv/L)/TAR(mmolTroloxequiv/L) [14]. Malondialdehyde (MDA), a lipid peroxidation marker, was measured using thiobarbituric acid and the results were expressed as nmol/mL of serum [15]. Total thiols (SH) were estimated using Ellman’s reagent and the results were expressed as mmol GSH/mL [16]. All serum spectrophotometric measurements were performed using a Jasco V-530 UV-Vis spectrophotometer (Jasco International Co. Ltd., Tokyo, Japan).

#### 2.3.4. Statistical Analysis

All results were expressed as mean ± standard deviation (SD) whenever data were normally distributed. Experimental groups were compared by using the one-way ANOVA test and the post hoc Bonferroni–Holm test. The correlations analysis between each group parameter was performed with the Pearson test. A *p* < 0.05 was considered statistically significant. The statistical analysis was performed by using IBM SPSS Statistics, version 20 (SPSS Inc., Chicago, IL, USA).

## 3. Results

### In Vivo Anti-Inflammatory and Antioxidative Effect

The anti-inflammatory activity of FLAX was evaluated by measuring serum levels of NOx, 3NT, and NF-kB. Inflammation increased NOx and 3NT too (*p* < 0.001). FLAX prophylactic and therapeutic treatment lowered NOx concentration in a dose dependent manner too, but the therapeutic treatment was more efficient than the prophylaxy. As compared to diclofenac, FLAX inhibitory activity on nitric oxide and 3NT production was better. In the serum of the INFLAM group, NFkB was significantly higher than in CONTROL (*p* < 0.001). Prophylactic and therapeutic treatments with FLAX caused a very significant reduction of NFkB in a dose dependent way, FLAX100 having the best inhibitory activity. FLAX effect on NFkB was better than the effect of diclofenac (Table 1).

Oxidative stress markers of the INFLAM animals were significantly changed as compared to the CONTROL. TAC and SH decreased (*p* < 0.001), and TOS, OSI and MDA increased (*p* < 0.001). The prophylactic and therapeutic treatments increased TAC, the prophylactic one having a better antioxidant effect. Diclofenac increased TAC as did FLAX100 in the prophylactic administration (Table 2).

TOS and OSI were reduced by FLAX100 and FLAX50 in both prophylactic and therapeutic administration, but the therapeutic treatment had better inhibitory activity. Diclofenac had a lower effect on TOS and OSI as compared to FLAX100 and FLAX50 in both treatments plans, prophylactic and therapeutic (Table 2). FLAX effects on TOS, OSI, NOx, 3NT, and NF-κB were correlated (*R^2^* = 0.98–0.68).

MDA increased after inflammation induction (*p* < 0.001). FLAX prophylactic treatment reduced MDA in a dose dependent way, FLAX100 being the most efficient. In the therapeutic plan, only FLAX100 and FLAX50 lowered MDA, and these effects were smaller than in the prophylactic plan. Diclofenac effect was comparable with FLAX100 and FLAX25 in the prophylactic plan, and FLAX100 in the therapeutic plan (Table 2).

SH was significantly reduced by the inflammation (*p <* 0.001). FLAX100 and FLAX50 improved SH in a concentration dependent way, and the prophylactic administration showed a better effect than the therapeutic one. Diclofenac effect was comparable with Flax50 as prophylactic administration, and FLAX100 as therapeutic treatment (Table 2).

## 4. Discussion

The results of this study provide scientific knowledge on the anti-inflammatory and antioxidant activities of FLAX in experimental acute inflammation.

In the last years, inflammation was a major target for the research areas. Today inflammation research has two therapeutic directions. The first is to find if anti-inflammatory drugs are efficient in one particular disease, and could be useful in other diseases too. Secondly, to find natural compounds with anti-inflammatory and antioxidant activity. According to the World Health Organization, 80% of the world’s population utilized traditional medicine for their primary health care [17]. Herbal extracts and their active substances can be included in the daily diet as part of preventive medicine, or as anti-inflammatory and antioxidant therapeutic supplements. Due to the side effects of existing anti-inflammatory drugs, these natural supplements became more popular [10,17].

Currently, many new molecules and their mechanisms were discovered in the studies of plants used in traditional phytotherapy [18]. It was found that natural products’ antioxidant activities are due to the flavonoids and the phenolic content, trace metals such as Cu, Zn, Mg, Mn, and Se, tocopherols, carotenoids, and ascorbic acid [17,19]. Previous phytochemical analysis of FLAX found important quantities of lignans and hydroxybenzoic acid, and lower quantities of gallic acid, syringic acid, and hydroxycinnamic acid derivatives. All these compounds have antioxidant, anti-inflammatory, and antineoplastic activities [20,21,22,23,24,25]. Together, these data suggested that FLAX in vivo anti-inflammatory and antioxidant activity may be significant too.

The anti-inflammatory agents act at the level of systemic mediators, at the cellular level on the production of inflammatory mediators (NO, iNOS, PGE2, COX-2, TNF-α, and IL-6), and molecular level on the activation of transcription factors (NF-κB) [5].

NO is a small molecule, it penetrates rapidly across cell membranes, and it is rapidly oxidized [6,26]. It can act as a signaling molecule, a toxin, an oxidant, and a potential antioxidant. As a signaling molecule, it is involved in vasodilatation and neurotransmission. NO is toxic for the pathogens, in high concentrations or when it reacts with superoxide (O_2_•^−^) forming peroxynitrite (ONOO−), an oxidizing and nitrating agent [27]. At low concentrations, NO• can be an antioxidant because it can block chain reactions during lipid peroxidation [26,28]. In the INLFAM group, NOx increased significantly, and both FLAX treatments, prophylactic and therapeutic lowered it in a dose dependent way, with FLAX100 having the best inhibitory effect. FLAX effect on NO synthesis was better in the therapeutic administration that in the prophylactic one. Moreover, the therapeutic FLAX administration was as efficient as diclofenac on NOx reduction. As chronic exposure to RNS is associated with inflammatory diseases and cancer [6], the reduction of Nox by FLAX is an important activity.

Peroxynitrite is an oxidizing and nitrating agent. Nitration of tyrosine residues of proteins generates 3NT which is widely used as a biomarker for inflammation induced oxidative and nitrosative stress. Nitration of structural proteins can disrupt structural assembly, and nitration of signaling molecules or transcription factors can alter proteins’ physiological function, both processes having major pathological consequences. Peroxynitrite also mediates calcium dependent mitochondrial dysfunction and cell death via activation of calpains [3,27]. The 3NT levels were increased by inflammation induction, and FLAX treatment lowered them. The therapeutic administration of FLAX had a better Inhibitory effect on 3NT than the prophylactic plan, and both plans’ effect was dose dependent, with the higher dose having the best anti-inflammatory activity.

The oxidative stress definition states that it is „an imbalance between oxidants and antioxidants in favor of the oxidants, leading to a disruption of redox signaling and control and/or molecular damage” [29]. The consequences can be proteins, lipids, DNA damage, signal transduction disruption, mutation, and cell death. Turpentine-induced inflammation increased ROS production markers TOS and OSI, and only FLAX100 and FLAX50 treatments decreased them significantly. As for NOx and 3NT, the therapeutic plan was more efficient than the prophylactic treatment on ROS lowering activity, and these effects were as important as for diclofenac.

A major pathway for oxidative damage is hydroxyl free radical (OH∙) induced lipid peroxidation and DNA hydroxylation. The consequences of lipid peroxidation are a decrease in membrane fluidity, an increase in the leakiness of the membrane, and damage to the membrane proteins, thereby inactivating receptors, enzymes, and ion channels [30]. The serum marker of lipid peroxidation, MDA, was reduced by FLAX treatments, but the prophylactic plan had better inhibitory activity than the therapeutic plan. Only FLAX100 had a better lowering effect on MDA than diclofenac.

Enzymatic and nonenzymatic antioxidants act by neutralizing reactive species or by breaking the chain reactions. In addition, some proteins (transferrin, ceruloplasmin, albumin) that bind transition metal ions have antioxidant roles too by blocking these metal ions to react with H_2_O_2_ [30]. The general antioxidant activity was evaluated by serum TAC. FLAX treatments increased TAC, the effect being better in the prophylactic plan than in the therapeutic administration, and both plans had a lower antioxidant effect than diclofenac. Similar results were found by measuring SH. Even when the results suggested a good antioxidant activity for FLAX, the administration of antioxidant compounds has to be performed carefully because antioxidants may be useful or harmful depending on the situation [30]. Several explanations have been proposed in order to explain the failure of antioxidants in human diseases. One explanation is that oxidative stress is the consequence not the cause of the disease, and so the antioxidants are not enough [31]. Another explanation is that in some situations, like in cancer, ROS are needed for cancer cells apoptosis induction, and antioxidants can be pro-tumorigenic [32]. A third possibility is that the oxidative stress markers were not properly selected because there is no consensus which are the limits of the desirable and/or undesirable levels of oxidative stress [30].

Further, ROS and RNS can trigger intracellular signaling cascade that increases proinflammatory gene expression. It has been found that oxidative stress, mostly H_2_O_2_, activate NF-κB and antioxidants block NF-κB activation. The nuclear factor-κB NF-κB is an important transcription factor in inflammation, because it regulates the gene expression of proinflammatory mediators, and antioxidant enzymes [3]. Therefore, NF-κB would be a good target against inflammatory diseases. In the INFLAM animals NF-κB increased significantly, and FLA treatments lowered NF-κB expression in a dose dependent way in both treatment plans. The therapeutic FLAX administration had better inhibitory activity, which was correlated to ROS and RNS concentrations.

## 5. Conclusions

Due to the phytochemical composition consisting of substances with anti-inflammatory and antioxidant effects, FLAX prophylactic and therapeutic administration had a good inhibitory effect in turpentine-induced acute inflammation. FLAX anti-inflammatory activity was better in the therapeutic plan than in the prophylactic one, and consisted of NO, 3NT, and NF-κB reduction in a dose dependent way. An important observation is that FLAX’s therapeutic anti-inflammatory effect was better than that of diclofenac. When analyzing antioxidant activity, FLAX had different effects on ROS and antioxidants. ROS was reduced better in the therapeutic administration, and TOS and SH were increased more by the prophylactic administration. FLAX100 had the best anti-inflammatory and antioxidant activity, and it was more efficient than diclofenac. Taking together, these results suggest that the FLAX100 prophylaxis increase the antioxidant activity, and FLAX therapy has anti-inflammatory and antioxidant activities by reducing NF-κB, RNS, and ROS. Future studies will have to verify these effects on human subjects.

## Figures and Tables

**Table 1 medicina-58-00582-t001:** Serum anti-inflammatory markers.

	NOx (ng/mL)		3NT (ng/mL)		NFkB (ng/mL)	
CONTROL	39.355	±4.609	2.169	±0.282	28.429	±3.937
INFAM	53.555	±5.346 ^###^	6.510	±1.475 ^###^	76.131	±8.612 ^###^
FLAX100/INFAM	35.701	±3.871 ***	3.506	±1.095 ***	39.448	±2.426 ***
FLAX50/INFAM	37.824	±6.201 **	3.728	±0.997 ***	50.934	±3.070 **
FLAX25/INFAM	39.234	±8.150 **	4.384	±1.869 *	52.539	±8.323 *
INFLAM/FLAX100	27.108	±6.554 ***	3.420	±1.059 ***	26.841	±1.208 ***
INFLAM/FLAX50	29.464	±4.081 ***	3.510	±0.989 ***	37.885	±3.355 ***
INFLAM/FLAX25	31.788	±5.157 ***	3.977	±1.278 ***	39.153	±2.840 ***
INFAM/DICLO	31.550	±7.284 ***	4.510	±2.003 **	40.771	±4.603 **

NOx–nitrites and nitrates; 3NT—3-nitrothyrosine; NFkB—nuclear factor kB; CONTROL—negative control; INFLAM—inflammation; FLAX100/INFLAM—flaxseed extract 100% prophylaxy; FLAX50/INFLAM)—flaxseed extract 50% prophylaxy; FLAX25/INFLAM—flaxseed extract 25% prophylaxy; INFLAM/FLAX100—flaxseed extract 100% treatment; INFLAM/FLAX50—flaxseed extract 500% treatment; INFLAM/FLAX25—flaxseed extract 25% treatment; INFLAM/DICLO—treatment with diclofenac; ^#^ statistical significance vs. CONTROL; * statistical significance vs. INFLAM; ^###^ *p* < 0.001; * *p* < 0.05; ** *p* < 0.01; *** *p* < 0.001.

**Table 2 medicina-58-00582-t002:** Serum oxidative stress markers.

	TAC (mmol Troloxequiv/L)		TOS (mmol Troloxequiv/L)		OSI		MDA (nM/L)		SH (mM/L)	
CONTROL	1.093	±0.001	5.181	±0.899	5.157	±0.626	2.496	±0.258	442.000	±45.778
INFAM	1.088	±0.001 ^##^	6.798	±0.712 ^###^	6.570	±0.621 ^###^	4.310	±0.130 ^###^	273.800	±35.654 ^###^
FLAX100/INFAM	1.094	±0.002 ***	5.368	±0.710 **	4.662	±1.432 ***	2.893	±0.354 ***	420.500	±51.675 ***
FLAX50/INFAM	1.092	±0.001 **	5.546	±1.085 **	5.061	±0.604 **	3.238	±0.411 **	313.000	±35.515 **
FLAX25/INFAM	1.091	±0.002 **	6.387	±0.709	5.269	±0.374 *	3.248	±0.245 **	289.333	±80.353
INFLAM/FLAX100	1.090	±0.002 **	4.008	±0.412 ***	3.633	±0.856 ***	3.173	±0.764 **	357.000	±28.107 ***
INFLAM/FLAX50	1.089	±0.002 *	4.295	±0.533 ***	4.128	±0.662 ***	3.364	±0.522 **	313.800	±35.344 **
INFLAM/FLAX25	1.091	±0.001 **	6.224	±0.819	5.749	±0.405 *	3.931	±0.215	287.400	±34.392
INFAM/DICLO	1.095	±0.002 ***	5.807	±0.586 **	5.272	±0.724 *	3.285	±0.262 **	360.000	±40.214 **

TAR—total antioxidant reactivity; TOS—total oxidative status; OSI—oxidative stress index; MDA—malondialdehyde; SH—total thiols; CONTROL—negative control; INFLAM–inflammation; FLAX100/INFLAM—flaxseed extract 100% prophylaxy; FLAX50/INFLAM)—flaxseed extract 50% prophylaxy; FLAX25/INFLAM–flaxseed extract 25% prophylaxy; INFLAM/FLAX100—flaxseed extract 100% treatment; INFLAM/FLAX50—flaxseed extract 500% treatment; INFLAM/FLAX25—flaxseed extract 25% treatment; INFLAM/DICLO—treatment with diclofenac; ^#^ statistical significance vs. CONTROL; * statistical significance vs. INFLAM; ^##^ *p* < 0.001; ^###^ *p* < 0.001; * *p* < 0.05; ** *p* < 0.01; *** *p* < 0.001.

## Data Availability

Data are going to be available in the PhD thesis of the first author after the public presentation at Iuliu Hatieganu University of Medicine and Pharmacy.

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
