# Peer review of "Flaxseed Ethanol Extract Effect in Acute Experimental Inflammation"

_medicina, 2022, doi:10.3390/medicina58050582_

Round 1

Reviewer 1 Report

The study evaluated the prophylactic and therapeutic anti-inflammatory and antioxidant effects of flaxseed ethanol extract in acute case of experimental inflammation. Here are some observations to be made in the paper. 

-On line 86, note the space between μm and Retsch;

-Standardize mL or ml, since the text has both forms;

- Reduce the paragraph in the discussion from line 194 to 209; 

Reviewer 2 Report

Title: Flaxseed Ethanol Extract Effect in Acute Experimental Inflammation

The present study investigated the preventive and therapeutic anti-inflammatory and antioxidant potential of flaxseed extract in an experimental turpentine oil induced acute inflammation.
Overall, the study is well-designed and well-written, the methods are suitable and the obtained results are clearly presented and discussed.  Moreover, the conclusions have been appropriately pointed out.
The content of this manuscript matches well with the journal’s purpose, but some clarifications are you can do:

line 89: should be: 70% ethanol (1/1.2 g/mL)

line 142: should be: A p<0.05 was considered

line 148, 153: should be: (p<0.001). = should be italic - correct all article

line 173: r - is it pearson correlation? if yes the "R2" should be italic 

Author Response

This manuscript is a resubmission of an earlier submission. The following is a list of the peer review reports and author responses from that submission.